# DRIK: DISTRIBUTION-ROBUST INDUCTIVE KRIGING WITHOUT INFORMATION LEAKAGE

## ABSTRACT

Inductive kriging supports high-resolution spatio-temporal estimation with sparse sensor networks, but conventional training–evaluation setups often suffer from information leakage and poor out-of-distribution (OOD) generalization. We find that the common 2×2 spatio-temporal split allows test data to influence model selection through early stopping, obscuring the true OOD characteristics of inductive kriging. To address this issue, we propose a 3×3 partition that cleanly separates training, validation, and test sets, eliminating leakage and better reflecting real-world applications. Building on this redefined setting, we introduce DRIK, a Distribution-Robust Inductive Kriging approach designed with the intrinsic properties of inductive kriging in mind to explicitly enhance OOD generalization, employing a three-tier strategy at the node, edge, and subgraph levels. DRIK perturbs node coordinates to capture continuous spatial relationships, drops edges to reduce ambiguity in information flow and increase topological diversity, and adds pseudo-labeled subgraphs to strengthen domain generalization. Experiments on six diverse spatio-temporal datasets show that DRIK consistently outperforms existing methods, achieving up to $12.48\%$ lower MAE while maintaining strong scalability.

## 1   INTRODUCTION

Sensors are widely used to monitor traffic flow (Kong et al., 2024), air quality (Yu et al., 2025), and solar energy production (Jebli et al., 2021), among other applications. However, their high deployment costs often limit sensor density and prevent comprehensive coverage of large areas (Liang et al., 2019; Seo et al., 2017). Inductive kriging provides a promising solution by estimating values at unsensed locations using data from existing sensors (Wu et al., 2021a; Zheng et al., 2023; Xu et al., 2025). Kriging models can generate high-resolution spatio-temporal estimates, improving accuracy while reducing the deployment and maintenance demands of large-scale sensor networks.

### 1.1   REDEFINING THE INDUCTIVE KRIGING SETTING

The standard training and evaluation protocol for inductive kriging (Wu et al., 2021a) generally involves three steps, as shown in Figure 1 (a): (1) The complete spatio-temporal dataset $\boldsymbol{X} \in \mathbb{R}^{N \times T}$ is split along both temporal and spatial dimensions, creating separate training and test periods as well as training and test nodes. This produces a 2×2 partition, with the final training and test sets drawn from diagonally opposite sections. (2) During training, the model is fitted to the training set, typically using masking and reconstruction techniques. (3) During testing, all training nodes from the test period are used to predict values at the test nodes.

A key limitation of this approach stems from the widespread use of early stopping during model training (Zheng et al., 2023). In the current protocol, model selection relies on the lowest loss achieved on the test set, which introduces data leakage by allowing test-set information to influence model development. Some studies have attempted to address this issue by adding a validation period along the temporal dimension, resulting in a 2×3 split (Xu et al., 2025; Zhu et al., 2025) (Figure 1 (b)). However, this adjustment still fails to prevent leakage of spatial information.

We propose a revised inductive kriging protocol that mitigates data leakage through a structured 3×3 partitioning scheme, as illustrated in Figure 1 (c): (1) The dataset is divided along the temporal

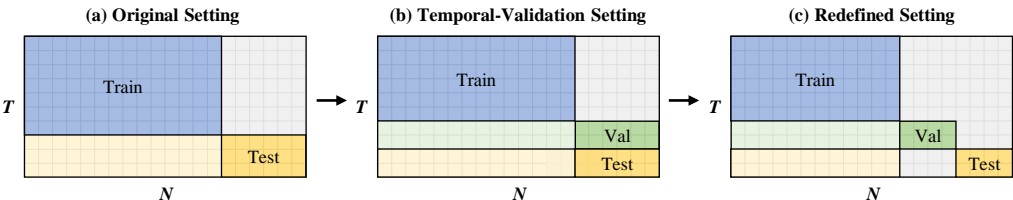

Figure 1: Comparison of inductive kriging settings. For better visualization, the matrix has been transposed. Blue, green, and yellow indicate the training, validation, and test sets, respectively. Light green and light yellow represent observed data from the validation and test periods used for prediction, while gray denotes data that remain unused throughout the process.

dimension into training, validation, and test periods, and along the spatial dimension into separate sets of training, validation, and test nodes. The training, validation, and test sets occupy the diagonal of the 3×3 grid. (2) During training, only the training set is used to fit the model. (3) During validation, all training nodes from the validation period are used to predict values at the validation nodes, and the model is selected based on the lowest validation loss. (4) During testing, all training nodes from the test period are used to predict values at the test nodes.

## 1.2 Challenges and Proposed Solution For the New Setting

Under the new setting, the key out-of-distribution (OOD) property of inductive kriging becomes clear, while previously it was underestimated due to information leakage (Wu et al., 2022b; Li et al., 2025). Differences between the training data and the kriging data induce a distinct distribution shift across both time and space—particularly in the spatial dimension—where the shift is substantial and cannot be ignored. This arises because current inductive kriging models encode spatial information with graphs whose topology is fixed during training, yet adding new nodes during kriging inevitably alters both graph density and topology, creating a significant challenge for model generalization (Xu et al., 2025). To overcome this challenge, we propose DRIK, an approach that mitigates the OOD problem and enables distribution-robust inductive kriging without information leakage.

DRIK leverages the unique training characteristics of inductive kriging to enhance distribution robustness through a three-tier strategy at the node, edge, and subgraph levels (Figure 2). At the node level, each node is perturbed within a limited range of its true coordinates and treated as a node domain, introducing controlled noise that captures the continuous spatial relationships required for kriging but unevenly discretized by graphs. At the edge level, outgoing edges of masked nodes, along with all edges between them, are removed to reduce ambiguity in information propagation and increase topological diversity. At the subgraph level, validation nodes are added during training without using their data; pseudo-labels are first generated through kriging, after which the masking and kriging steps are repeated. This process further strengthens the model's ability to generalize to unseen domains. Extensive experiments demonstrate that the model achieves superior performance and stronger generalization across multiple datasets.

## 1.3 Contributions

Our contributions can be summarized as follows:

- We redefine the inductive kriging setting by redesigning the division of training, validation, and test sets, eliminating the information leakage found in prior task designs and aligning the setting more closely with real-world kriging applications.

- We identify distribution shift as a key factor limiting the performance of inductive kriging models. We demonstrate the OOD property of inductive kriging and introduce a three-level strategy—node, edge, and subgraph—to enhance distribution robustness.

- We conducted extensive experiments on six spatio-temporal datasets spanning three categories. Our approach consistently outperformed existing methods, reducing error by up to 12.48%. It also showed stronger generalization, evidenced by a lower test-to-validation MAE ratio across all datasets.

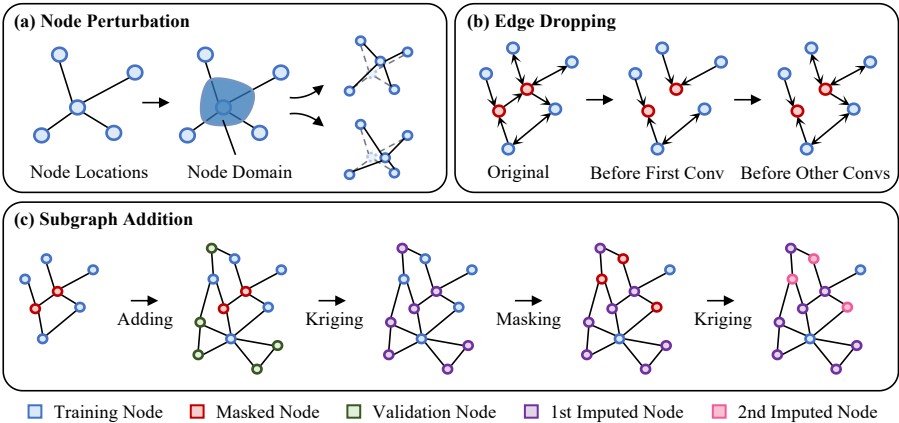

Figure 2: Overview of DRIK. (a) Illustration of node-level strategy. Perturb nodes within a limited range of their true coordinates to create node domains. (b) Illustration of edge-level strategy. Drop the outgoing edges of masked nodes and all edges between them. (c) Illustration of subgraph-level strategy. Add validation nodes in advance, generate pseudo-labels through an initial kriging, then perform a second masking and kriging.

## 2 PRIOR WORKS

**Inductive Kriging.** Kriging is a widely used geostatistical technique for spatial interpolation, where the value at an unsampled location is predicted from observations at nearby sites (Krige, 1951; Oliver & Webster, 1990). Kriging can be classified as inductive, which predicts entirely unknown nodes, or transductive, which resembles missing-value imputation (see Appendix A.1 for details). Graph neural networks (GNNs) have become the dominant approach for inductive kriging. Pioneering methods such as KCN (Appleby et al., 2020) and IGNNK (Wu et al., 2021a) were the first to apply GNNs to kriging, achieving significant improvements over traditional approaches (Zhou et al., 2012; Bahadori et al., 2014). Building on these, subsequent models including SATCN (Wu et al., 2021b), LSJSTN (Hu et al., 2023), INCREASE (Zheng et al., 2023), IAGCN (Wei et al., 2024), and DBGNN (Zhu et al., 2025) have further enhanced the integration of temporal information, spatial information, and additional covariates, reporting improved performance. KITS (Xu et al., 2025) identified the graph gap—the training graph is much sparser than the inference graph containing all observed and unobserved nodes—and sought to mitigate it by replacing the usual decrement training strategy with an increment training strategy. Despite these advances, existing methods continue to exhibit information leakage, which understates the severity of the OOD problem, particularly in the spatial dimension.

**OOD Generalization on Graphs.** OOD generalization on graphs (Wu et al., 2022b; Li et al., 2025) remains a persistent challenge in graph machine learning, as real-world graph data often exhibit distribution shifts. Data augmentation has emerged as an effective strategy for enhancing model robustness under such shifts. These methods can be categorized into three types: structural, feature-based, and hybrid augmentations. Structural augmentation modifies graph topology to expose models to varied connectivity patterns, as seen in GAug (Zhao et al., 2021), MH-Aug (Park et al., 2021), and KDGA (Wu et al., 2022a). Feature-based augmentation perturbs node attributes to promote invariance to feature noise, exemplified by GRAND (Feng et al., 2020), FLAG (Kong et al., 2022), and LA-GNN (Liu et al., 2022b). Hybrid methods combine both structural and feature manipulations, such as in GraphCL (You et al., 2020), GREA (Liu et al., 2022a), and AIA (Sui et al., 2023). For more methods beyond data augmentation, we refer readers to Appendix A.2. Despite these advances, few methods explicitly address OOD generalization in the context of inductive kriging, which involves distinctive graph characteristics such as spatially embedded nodes, intrinsically masked nodes, and underutilized substructures. These properties remain underexplored in current augmentation strategies. Leveraging them could significantly improve generalization performance in inductive spatio-temporal kriging.

## 3 METHODOLOGY

In this section, we first define the problem to be addressed and introduce the spatio-temporal graph convolution required for kriging, along with the associated OOD challenge. Building on these concepts and the characteristics of inductive kriging, we then present DRIK, a method that improves OOD generalization by adopting targeted strategies at three levels: node, edge, and subgraph.

### 3.1 PROBLEM DEFINITION

**Terminology.** Consider $\boldsymbol{X}^o_{T-t:T} \in \mathbb{R}^{N_o \times t}$, which represents the observed values of $N_o$ nodes over $t$ time intervals. Following the approach of Wu et al. (2021a) and Xu et al. (2025), we construct a graph on these observed nodes using the Gaussian kernel function to establish the edges. The adjacency matrix of this graph is denoted as $\boldsymbol{A}_o \in [0,1]^{N_o \times N_o}$. The goal of inductive kriging is to predict the values of $N_u$ unobserved nodes—whose values are unknown until inference—using both $\boldsymbol{X}^o_{T-t:T}$ and the graph that incorporates the observed and unobserved nodes. The primary distinction from previous studies lies in the dataset partitioning, as discussed in Section 1.1.

**Spatio-Temporal Graph Convolution (STGC).** STGC serves as the core component of the kriging model, aggregating spatio-temporal features from neighboring nodes via graph convolution (Cini et al., 2022; Xu et al., 2025). Let the input features be $\boldsymbol{Z}_i \in \mathbb{R}^{N_o \times D}$, where $i$ is the time interval $T_i$ and $D$ is the feature dimension. To capture temporal context, $\boldsymbol{Z}_i$ is concatenated with features from the previous and next $m$ intervals, yielding $\boldsymbol{Z}_{i-m:i+m} \in \mathbb{R}^{N_o \times (2m+1)D}$. Spatial aggregation then uses the training graph with adjacency matrix $\boldsymbol{A}_o$, where the diagonal is masked ($\boldsymbol{A}_o^-$) to remove self-loops. Formally, STGC can be written as:

$$\boldsymbol{Z}_i^{(l+1)} = \mathrm{FC}\Big(\mathrm{GC}\big(\boldsymbol{Z}_{i-m:i+m}^{(l)}, \boldsymbol{A}_o^-\big)\Big), \tag{1}$$

where $(l)$ and $(l+1)$ are layer indices, $\mathrm{FC}(\cdot)$ is a fully connected layer, and $\mathrm{GC}(\cdot)$ is the inductive graph convolution layer.

**OOD Problem.** We train on the observed subgraph $\mathcal{G}_o = (V_o, \boldsymbol{A}_o)$ and evaluate kriging on the enlarged graph $\mathcal{G} = (V_o \cup V_u, \boldsymbol{A})$ with previously unseen nodes $V_u$. The adjacency matrix is partitioned as

$$\boldsymbol{A} = \begin{pmatrix} \boldsymbol{A}_{oo} & \boldsymbol{A}_{ou} \\ \boldsymbol{A}_{uo} & \boldsymbol{A}_{uu} \end{pmatrix}, \qquad \boldsymbol{A}_{oo} = \boldsymbol{A}_o. \tag{2}$$

Consequently, empirical risk minimization fits

$$\mathcal{R}_{\mathrm{train}}(\theta) = \mathbb{E}_{(\boldsymbol{X}^o, \boldsymbol{A}_{oo}) \sim P_o} \left[ \ell\left(f_\theta(\boldsymbol{X}^o; \boldsymbol{A}_{oo}), \boldsymbol{Y}^o\right) \right], \tag{3}$$

whereas evaluation on the validation or test set measures

$$\mathcal{R}_{\mathrm{eval}}(\theta) = \mathbb{E}_{(\boldsymbol{X}^o, \boldsymbol{A}) \sim P_{ou}} \left[ \ell\left(f_\theta(\boldsymbol{X}^o; \boldsymbol{A}), \boldsymbol{Y}^u\right) \right]. \tag{4}$$

Here, $f_\theta$ is the STGC stack in Eq. 1. The normalized propagation operator changes from $\hat{\boldsymbol{A}}_{oo}$ to $\hat{\boldsymbol{A}}$, which introduces both a degree-matrix and spectrum shift. The target also shifts from masked reconstruction on observed nodes $\boldsymbol{Y}^o$ to extrapolation on unseen nodes $\boldsymbol{Y}^u$. As a result, intermediate features undergo both structural and covariate shifts. From a Gaussian-process / kriging perspective the conditional mean is

$$\boldsymbol{\mu}_{u|o} = \boldsymbol{K}_{uo} \, \boldsymbol{K}_{oo}^{-1} \, \boldsymbol{X}^o, \tag{5}$$

where $\boldsymbol{K}$ denotes the kernel matrix. Training only approximates $\boldsymbol{K}_{oo}^{-1}$ via masked self-supervision on $V_o$, whereas evaluation relies on $\boldsymbol{K}_{uo}$, which encodes spatial relations between $V_u$ and $V_o$; this mismatch yields a natural OOD setting. Additional theoretical details appear in Appendix B.

### 3.2 DISTRIBUTION-ROBUST INDUCTIVE KRIGING

We instantiate DRIK as a three-pronged scheme acting on (i) node coordinates, (ii) masked-node connectivity, and (iii) train/validation subgraph composition (see Fig. 2). Concretely, DRIK perturbs each node within a geometry-aware node domain to partially restore spatial continuity, prunes ambiguous edges involving masked nodes to stabilize propagation, and exploits validation-node topology via a two-stage pseudo-labeling routine—without leaking validation measurements.

**Node Perturbation.** Inductive kriging constructs a graph from node coordinates and pairwise distances, effectively discretizing an underlying continuous spatial process. Because this graph is fixed once built, adding new nodes can substantially change its structure. To restore spatial continuity and improve generalization, we introduce node perturbation, which randomly shifts node locations during training.

Let the observed-node set be $V_o$, where each node $v \in V_o$ has coordinates $s_v \in \mathbb{R}^d$ (typically $d = 2$). Let $\mathcal{N}(v)$ denote the neighbor set of $v$ under the inductive-kriging graph (for example, a $k$-nearest-neighbor graph with a Gaussian kernel). We define the node domain $\mathcal{D}_v \subset \mathbb{R}^d$ as the convex hull of midpoints between $v$ and its neighbors:

$$\mathcal{D}_v = \mathrm{conv}\Big\{ m_{v,u} = s_v + \tfrac{1}{2}(s_u - s_v) \mid u \in \mathcal{N}(v)\Big\}, \tag{6}$$

where the scale of $\mathcal{D}_v$ is determined by inter-node distances $\|s_v - s_u\|$. The vertices of $\mathcal{D}_v$ are the midpoints between $v$ and its neighbors, and their convex hull forms the node domain.

During each training iteration $r$, all nodes—whether masked or unmasked—are perturbed by sampling

$$\tilde{s}_v^{(r)} \sim \mathrm{Unif}(\mathcal{D}_v), \tag{7}$$

followed by rebuilding the adjacency matrix using a kernelized, row-normalized $k$-nearest-neighbor graph:

$$\tilde{\boldsymbol{A}}_o^{(r)}(v, u) = \frac{\exp\big(-\|\tilde{s}_v^{(r)} - \tilde{s}_u^{(r)}\|^2/\sigma^2\big)}{\sum_{u' \in \mathrm{kNN}_k(v)} \exp\big(-\|\tilde{s}_v^{(r)} - \tilde{s}_{u'}^{(r)}\|^2/\sigma^2\big)} \cdot \mathbf{1}\big\{u \in \mathrm{kNN}_k(v)\big\}, \qquad v, u \in V_o. \tag{8}$$

Here, $\sigma > 0$ is the Gaussian kernel bandwidth (length-scale) that controls how rapidly the edge weight decays with distance. This continuity-aware perturbation exposes the model to a family of propagation operators $\{\tilde{\boldsymbol{A}}_o^{(r)}\}$, reducing sensitivity to graph discretization and improving robustness to unseen node geometries.

**Edge Dropping.** Let $\mathcal{M} \subseteq V_o$ be the set of masked training nodes in the current mini-batch, which provide self-supervised targets on $V_o$. We define the layer-wise edge-drop operator $\Phi^{(l)} : \mathbb{R}^{N_o \times N_o} \to \mathbb{R}^{N_o \times N_o}$, applied just before the $l$-th graph convolution. This operator is central to a principled masking mechanism in inductive kriging, ensuring that message passing reflects only reliable information while progressively incorporating masked nodes into the learning process.

Before the first convolution ($l = 0$), we remove all edges between masked nodes as well as all outgoing edges from masked nodes,

$$\big(\Phi^{(0)}(\tilde{\boldsymbol{A}}_o)\big)_{vu} = \tilde{\boldsymbol{A}}_{o,vu} \, \mathbf{1}\{v \notin \mathcal{M}\} \, \mathbf{1}\{\neg(v \in \mathcal{M} \wedge u \in \mathcal{M})\}. \tag{9}$$

This initial pruning is crucial. At the outset, masked nodes have no reliable feature representations because their true labels are intentionally hidden for self-supervised learning. If their outgoing edges were retained, these nodes could inject uninformative or misleading signals into neighboring unmasked nodes. Furthermore, links between masked nodes would allow mutual reinforcement of uninitialized features, amplifying noise during the very first aggregation step and degrading the quality of propagated information.

For subsequent convolutions ($l \geq 1$), the dropping rule is relaxed to remove only edges between masked nodes,

$$\big(\Phi^{(l)}(\tilde{\boldsymbol{A}}_o)\big)_{vu} = \tilde{\boldsymbol{A}}_{o,vu} \, \mathbf{1}\{\neg(v \in \mathcal{M} \wedge u \in \mathcal{M})\}. \tag{10}$$

By this stage, masked nodes already encode partially aggregated and more reliable feature signals derived from earlier rounds of message passing. Consequently, their outgoing edges to unmasked neighbors are reinstated to allow normal propagation, while masked-to-masked connections remain suppressed to avoid circular error accumulation and to maintain stability during the kriging of unknown values. This progressive relaxation allows masked nodes to gradually participate in the graph convolution while still guarding against feedback loops that could compromise prediction accuracy.

Unlike conventional random edge-drop strategies, our edge-dropping method is a task-aware mechanism aligned with the inductive-kriging objective. By selectively controlling edge participation, it reduces early-stage ambiguity, limits spurious correlations, and enhances both training stability and generalization to unseen graph structures.

**Subgraph Addition.** In the redefined kriging setting, we exploit the validation nodes by revealing only their induced topology while masking their measurements (reserved for validation loss and model selection). This topology-only augmentation regularizes the operator and enhances generalization to unseen domains without leaking validation information into training. Let $V_{\text{tr}}$ and $V_{\text{val}}$ denote the training and validation node sets, respectively, and let $\widetilde{\boldsymbol{A}}_\cup$ be the adjacency matrix constructed in the same way as for the training graph but over the union of nodes. The augmented graph is therefore

$$\mathcal{G}_\cup = \big(V_{\text{tr}} \cup V_{\text{val}},\ \widetilde{\boldsymbol{A}}_\cup\big). \tag{11}$$

We perform two kriging passes on $\mathcal{G}_\cup$ using disjoint training masks. In the first pass, we randomly select a masked set $\mathcal{M}_1 \subset V_{\text{tr}}$ and mask all nodes in $V_{\text{val}}$. Using only the available training measurements $\boldsymbol{X}$ as input features, we compute

$$\big(\widehat{\boldsymbol{Y}}_{V_{\text{tr}}}^{(1)}, \widehat{\boldsymbol{Y}}_{V_{\text{val}}}^{(1)}\big) = f_\theta\Big(\boldsymbol{X};\ \Phi^{(0:L-1)}(\widetilde{\boldsymbol{A}}_\cup)\Big), \qquad \widetilde{\boldsymbol{Y}}_{V_{\text{val}}} := \text{stopgrad}\Big(\widehat{\boldsymbol{Y}}_{V_{\text{val}}}^{(1)}\Big), \tag{12}$$

so that

$$\frac{\partial \widetilde{\boldsymbol{Y}}_{V_{\text{val}}}}{\partial \theta} = 0, \tag{13}$$

ensuring that no gradient flows back from the validation predictions.

In the second pass, we resample another masked set $\mathcal{M}_2 \subset V_{\text{tr}}$ with $\mathcal{M}_2 \cap \mathcal{M}_1 = \varnothing$ and clamp the validation-node features to $\widetilde{\boldsymbol{Y}}_{V_{\text{val}}}$:

$$\boldsymbol{X}\big|_{V_{\text{val}}} \leftarrow \widetilde{\boldsymbol{Y}}_{V_{\text{val}}}, \qquad \widehat{\boldsymbol{Y}}_{V_{\text{tr}}}^{(2)} = f_\theta\Big(\boldsymbol{X};\ \Phi^{(0:L-1)}(\widetilde{\boldsymbol{A}}_\cup)\Big). \tag{14}$$

Training minimizes MAE over all training nodes masked in either pass:

$$\mathcal{L}_{\text{DRIK}} = \frac{1}{|\mathcal{M}_1 \cup \mathcal{M}_2|} \sum_{v \in \mathcal{M}_1 \cup \mathcal{M}_2} \big|\widehat{\boldsymbol{Y}}_v^{(\pi(v))} - \boldsymbol{Y}_v\big|, \qquad \pi(v) = \begin{cases} 1, & v \in \mathcal{M}_1, \\ 2, & v \in \mathcal{M}_2, \end{cases} \tag{15}$$

where $\boldsymbol{Y}_v$ represents the true measurement at node $v$. Only training nodes contribute to $\mathcal{L}_{\text{DRIK}}$, while the validation loss is computed separately on $V_{\text{val}}$ using their true measurements.

## 4 EXPERIMENTS

In this section, we conduct experiments to address the following research questions:

- **RQ1:** How does DRIK perform on inductive kriging tasks compared with baseline methods? Does it demonstrate advantages across different spatio-temporal datasets?
- **RQ2:** How can the degree of distribution shift in inductive kriging be measured? Can DRIK effectively mitigate the OOD problem?
- **RQ3:** How do the three levels of strategies in DRIK interact to achieve the final results? Does each module contribute meaningfully to overall performance?
- **RQ4:** How does DRIK's performance change as the degree of missingness varies? Is DRIK robust across different missing rates (e.g., under high missingness)?

### 4.1 EXPERIMENTAL SETUP

We begin by briefly outlining the datasets, baseline methods, and evaluation metrics. A more detailed description of the experimental settings is provided in Appendix C.

**Datasets & Splits.** We evaluate DRIK on six public datasets drawn from diverse real-world scenarios: two traffic datasets (METR-LA and PEMS-BAY) (Li et al., 2018), two solar-power datasets (NREL-AL and NREL-MD) (Bloom et al., 2016), and two air-quality datasets (AQI-36 and AQI) (Yi et al., 2016). Following Wu et al. (2021a), we randomly select 25% of sensors in each dataset as unobserved locations, with the remainder serving as observed locations. The training, validation, and test nodes account for 60%, 20%, and 20% of the observed locations, respectively. Along the

Table 1: Comparison of DRIK with existing methods on the inductive kriging task. The best results are highlighted in **bold**, while the second-best results are underlined. "Improvements" show the improvement of our DRIK over the best baseline.

| Method | METR-LA (207) | | | PEMS-BAY (325) | | | NREL-AL (137) | | |
|---|---|---|---|---|---|---|---|---|---|
| | MAE↓ | RMSE↓ | MAPE↓ | MAE↓ | RMSE↓ | MAPE↓ | MAE↓ | RMSE↓ | MAPE↓ |
| MEAN | 8.272 | 11.417 | 22.133 | 4.999 | 8.474 | 12.862 | 5.492 | 8.353 | 166.221 |
| OKriging | 7.294 | 10.277 | 18.896 | 4.874 | 8.266 | 12.412 | 7.960 | 10.580 | 406.106 |
| KNN | 7.987 | 12.370 | 19.820 | 5.678 | 10.431 | 14.087 | 7.962 | 10.582 | 410.155 |
| KCN | 7.190 | 12.470 | 23.983 | 4.676 | 9.253 | 13.514 | 4.541 | 6.697 | 155.001 |
| IGNNK | 5.801 | 8.914 | 15.581 | 3.445 | 6.067 | 8.378 | 4.531 | 6.619 | 160.523 |
| INCREASE | 5.992 | 9.198 | 16.854 | 3.599 | 6.850 | 9.457 | 5.524 | 7.950 | 116.402 |
| KITS | 5.666 | 8.981 | 15.096 | 3.410 | 6.445 | 8.602 | 4.532 | 6.510 | 177.941 |
| DRIK(Ours) | **5.197** | **8.101** | **13.154** | **3.218** | **5.840** | **7.728** | **3.966** | **6.357** | **81.963** |
| Improvements | **8.28%** | **9.12%** | **12.86%** | **5.63%** | **3.75%** | **7.76%** | **12.48%** | **2.35%** | **29.59%** |

| Method | NREL-MD (80) | | | AQI-36 (36) | | | AQI (437) | | |
|---|---|---|---|---|---|---|---|---|---|
| | MAE↓ | RMSE↓ | MAPE↓ | MAE↓ | RMSE↓ | MAPE↓ | MAE↓ | RMSE↓ | MAPE↓ |
| MEAN | 11.257 | 16.387 | 294.610 | 18.431 | 31.631 | 49.586 | 39.718 | 59.968 | 142.226 |
| OKriging | 11.947 | 16.455 | 703.908 | 16.003 | 28.744 | 42.670 | 23.827 | 39.846 | 85.340 |
| KNN | 11.953 | 16.464 | 706.322 | 14.727 | 26.800 | 37.737 | 18.376 | 32.490 | 52.270 |
| KCN | 10.961 | 17.032 | 173.269 | 21.963 | 36.647 | 57.988 | 21.012 | 35.111 | 61.017 |
| IGNNK | 11.011 | 17.308 | 195.432 | 20.138 | 33.993 | 69.964 | 16.315 | 29.448 | 44.635 |
| INCREASE | 10.282 | 16.271 | 147.958 | 16.963 | 32.854 | 41.619 | 16.034 | 29.862 | 43.268 |
| KITS | 11.601 | 17.589 | 444.394 | 19.600 | 34.668 | 76.466 | 16.068 | 29.791 | **39.033** |
| DRIK(Ours) | **10.151** | **16.163** | **95.635** | **13.443** | **25.550** | **28.433** | **15.364** | **28.437** | 40.180 |
| Improvements | **1.28%** | **0.66%** | **35.36%** | **8.71%** | **4.67%** | **24.65%** | **4.18%** | **3.43%** | **−2.94%** |

temporal dimension, following Xu et al. (2025), the training, validation, and test periods cover 70%, 10%, and 20% of the total time span.

**Baseline Methods & Evaluation Metrics.** We compare our method against several inductive kriging baselines, including Mean imputation, OKriging (Cressie & Wikle, 2015), K-nearest neighbors (KNN), KCN (Appleby et al., 2020), IGNNK (Wu et al., 2021a), INCREASE (Zheng et al., 2023) and KITS (Xu et al., 2025). We employ the Mean Absolute Error (MAE), Root Mean Square Error (RMSE), and Mean Absolute Percentage Error (MAPE) as evaluation metrics.

### 4.2 PERFORMANCE ON INDUCTIVE KRIGING (RQ1)

Table 1 presents the inductive-kriging results on six datasets. Additional experimental results, including analyses of model stability under different node divisions, are provided in Appendix D.1. From Table 1, we draw the following observations:

- **Obs 1: DRIK achieves superior performance across all datasets and metrics.** It outperforms existing methods in terms of MAE, RMSE, and MAPE. For example, on the NREL-AL dataset DRIK reduces MAE by 12.48% compared with the best baseline; on the AQI-36 dataset, it reduces MAPE by up to 24.65%; and on METR-LA, it achieves gains of 8.28%, 9.12%, and 12.86% in MAE, RMSE, and MAPE, respectively. These improvements stem from DRIK's three-tier strategy—node perturbation, edge dropping, and subgraph addition—which together enhance distributional robustness and alleviate OOD generalization issues.

- **Obs 2: DRIK demonstrates strong generalization across diverse application domains, with more pronounced advantages in complex scenarios.** The method delivers notable improvements on traffic, solar energy, and air quality datasets, reflecting its adaptability to varied data characteristics. In tasks with greater spatial heterogeneity, such as air quality and solar energy, DRIK achieves MAPE improvements of up to 35.36%, 29.59%, and 24.65% on the NREL-MD, NREL-AL, and AQI-36 datasets, respectively, underscoring its capacity to handle complex spatio-temporal distributions.

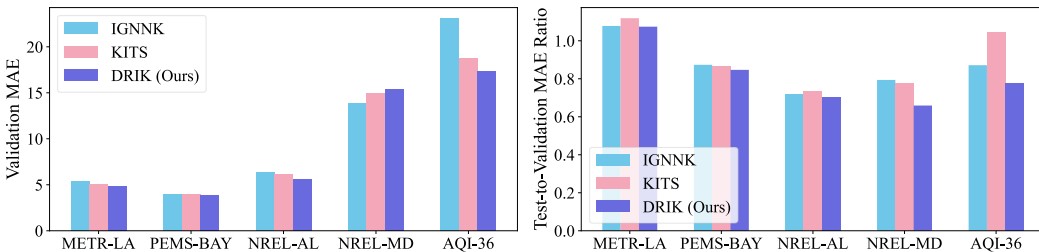

Figure 3: OOD property evaluation of IGNNK, KITS, and DRIK. A smaller test-to-validation MAE ratio indicates stronger generalization ability.

### 4.3 OOD Property Evaluation (RQ2)

To further verify the OOD property of inductive kriging and the OOD generalization capability of DRIK, we recorded the MAE values of three representative models—IGNNK, KITS, and DRIK—during training and evaluation. Specifically, we documented the lowest validation MAE, which was used to select the best model, and the MAE of that selected model on the test set (consistent with Table 1). Training MAE was not recorded because differences in masking strategies during training make cross-method comparisons unreliable. Figure 3 contains two subplots: one shows the validation MAE, and the other shows the ratio of test MAE to validation MAE. A smaller ratio indicates stronger generalization ability. Additional results are provided in Appendix D.2. Based on Figure 3, we draw the following observations:

- **Obs 3: The new 3×3 data-split format reveals the true differences in kriging accuracy and OOD capability among models.** Under the traditional 2×2 data-split setting, the validation MAE typically serves as the final evaluation metric, but the ranking of models based on validation MAE often differs from their test-set performance, highlighting the need for the new data-split setting to accurately assess model capability.
- **Obs 4: DRIK's performance gains across datasets primarily stem from its enhanced OOD generalization.** In terms of the test-to-validation MAE ratio, DRIK consistently outperforms the other two methods and shows clear advantages on the NREL-MD and AQI-36 datasets. A comparison of KITS and DRIK—both of which use the same STGC module—shows that DRIK does not always have a clear advantage in validation MAE (e.g., on NREL-MD and PEMS-BAY), yet its stronger OOD generalization leads to a significant overall performance improvement.

### 4.4 Ablation Study (RQ3)

Table 2 demonstrates the efficacy of each proposed module. M-0 denotes a configuration with no DRIK modules. According to Table 2 we can find that:

- **Obs 5: Single modules are not reliably effective, whereas combining modules yields consistent gains.** In isolation, NP improves MAE/RMSE but hurts MAPE, and SA degrades all metrics; even ED, the best single module, offers only modest gains. In contrast, pairwise combinations improve all metrics. This pattern suggests complementary inductive biases: NP enforces spatial continuity but can shift scale, ED suppresses noisy message passing yet has limited capacity alone,

Table 2: Component-wise ablation study. "NP", "ED", and "SA" denote Node Perturbation, Edge Dropping, and Subgraph Addition, respectively.

| Method | NP | ED | SA | MAE↓ | RMSE↓ | MAPE↓ |
|---|---|---|---|---|---|---|
| M-0 | | | | 6.090 | 9.372 | 16.274 |
| M-1 | ✓ | | | 5.781 | 9.115 | 16.682 |
| M-2 | | ✓ | | 5.713 | 8.994 | 15.459 |
| M-3 | | | ✓ | 6.419 | 10.092 | 17.056 |
| M-4 | ✓ | ✓ | | 5.368 | 8.335 | 14.685 |
| M-5 | ✓ | | ✓ | 5.589 | 8.604 | 13.465 |
| M-6 | | ✓ | ✓ | 5.674 | 9.031 | 15.252 |
| M-7 | ✓ | ✓ | ✓ | **5.197** | **8.101** | **13.154** |

and SA's pseudo-labels are unstable without structural regularization. When combined, these effects counterbalance—stabilizing topology and scale while enriching supervision—yielding robust, across-the-board improvements.

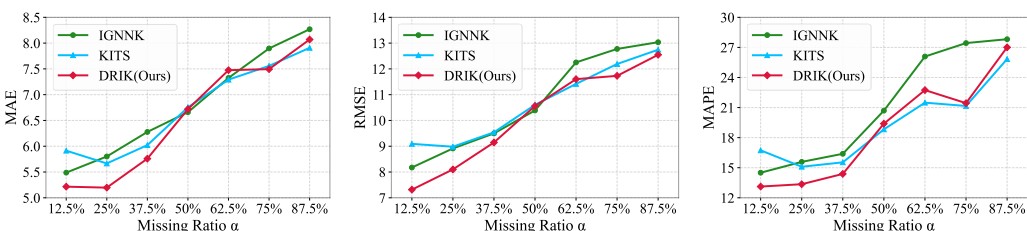

Figure 4: Comparisons with different missing ratios $\alpha$.

- **Obs 6: Using all three modules together yields the best overall robustness.** The full model (NP+ED+SA) achieves $5.197/8.101/13.154$, improving over the base by $-14.7\%$ MAE, $-13.6\%$ RMSE, $-19.2\%$ MAPE, and further surpassing the best two-module setting (NP+ED) by MAE $-3.2\%$, RMSE $-2.8\%$, MAPE $-10.4\%$. The flip of SA from harmful in isolation to net-positive in combinations indicates complementary supervision: pseudo-labels add value once edge ambiguity is reduced (ED) and spatial continuity is modeled (NP).

## 4.5 MISSING RATIO INFLUENCE ANALYSIS (RQ4)

As shown in Figure 4, we compare DRIK with other baseline methods as the missing ratio $\alpha$ increases from 12.5% to 87.5%, making the kriging task progressively harder. The results show that:

- **Obs 7: DRIK consistently achieves the best performance across datasets and metrics at low and medium missing ratios.** For example, when the missing ratio is below 50%, the MAE values at 12.5%, 25%, and 37.5% missing ratios are $5.216$, $5.197$, and $5.759$, representing reductions of $4.96\%$, $8.28\%$, and $4.35\%$, respectively, compared with the best baseline method.
- **Obs 8: DRIK remains competitive even at high missing ratios.** KITS employs an incremental training strategy (see Appendix A.3), which offers a clear advantage when the missing ratio is high. By contrast, DRIK adopts a decremental training approach, and additional edge dropping can further increase the likelihood of isolated nodes, hindering model training and potentially reducing accuracy. Even so, DRIK achieves performance comparable to KITS, indicating that subgraph addition effectively counteracts the node isolation caused by both decremental training and edge dropping.

## 5 LIMITATIONS & FUTURE DISCUSSION

While DRIK demonstrates strong capability and distributional robustness for inductive kriging, we also recognize its limitations. Under extreme conditions with very high missing ratios, DRIK can increase the likelihood of isolated nodes. Balancing generalization with the risk of excessive disconnection, for example through adaptive pruning based on local connectivity or spectral radius, remains an important direction for future work. Furthermore, our evaluation currently covers only traffic, photovoltaic, and air quality tasks, whereas kriging also has promising applications such as dynamical field reconstruction and regional subsidence estimation, which merit further exploration. These avenues offer opportunities to enhance both the applicability and scalability of our method.

## 6 CONCLUSIONS

In this work, we first identify the risk of information leakage in existing inductive kriging settings and propose a protocol that decouples data splitting across temporal and spatial dimensions, thereby revealing the OOD nature of inductive kriging. Building on this foundation, we introduce DRIK, a three-layer strategy comprising node perturbation, task-aware edge dropping, and subgraph addition to enhance OOD generalization. Extensive experiments on six datasets show that DRIK lowers MAE by up to $12.48\%$, achieves a markedly reduced test-to-validation MAE ratio, and delivers significant gains at low and medium missing rates while remaining competitive even at high missing rates.

## LARGE LANGUAGE MODEL USAGE STATEMENT

Large Language Models (LLMs) were used to assist in refining the manuscript's language and improving clarity. Their role was limited to polishing grammar, enhancing readability, and ensuring a consistent academic tone. All substantive ideas, analyses, and conclusions remain the authors' original work.

## REPRODUCIBILITY STATEMENT

The supplementary material contains the code and configuration files needed to reproduce the experiments and replicate the reported results. All datasets are publicly available, and download links are provided in the supplementary material. To ensure reproducibility and consistency across experiments and baselines, we use random number generators with fixed seeds to generate missing data.

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

APPENDIX

# A MORE DETAILED RELATED WORKS

## A.1 TRANSDUCTIVE KRIGING

Transductive kriging is a spatio-temporal interpolation method in which the set of unobserved locations must remain fixed during training. In essence, it performs spatio-temporal data imputation under the assumption that the prediction targets are known in advance. Existing approaches generally fall into three main categories. The first treats kriging as a missing-data completion problem and uses matrix or tensor factorization on a static tensor organized as location × time × variables (Zhou et al., 2012; Bahadori et al., 2014; Takeuchi et al., 2017; Deng et al., 2021). For example, GLTL (Bahadori et al., 2014) fills unobserved entries with zeros and applies tensor decomposition to estimate the missing values. This family of methods benefits from well-studied optimization techniques but often struggles to capture highly dynamic temporal patterns. A second category employs graph neural networks combined with recurrent architectures for spatio-temporal imputation (Cini et al., 2022; Marisca et al., 2022; Kong et al., 2023; Shen et al., 2023). Representative work such as GRIN (Cini et al., 2022) integrates message-passing mechanisms with Gated Recurrent Units (GRUs) to capture complex spatial dependencies and model temporal dynamics simultaneously, thereby reconstructing missing data at unobserved nodes more effectively than purely factorization-based models. A third direction frames transductive kriging as a generative modeling task, using probabilistic frameworks to impute missing data (Xu et al., 2020; Liu et al., 2023; Yun et al., 2023). Methods such as PriSTI (Liu et al., 2023) learn to generate plausible values for unobserved locations under uncertainty. Although effective within their respective settings, these methods remain inherently transductive: they assume that all unobserved nodes are predefined during training. Consequently, they cannot generalize to new or unseen locations without retraining, underscoring the need for inductive kriging approaches that can handle novel spatial contexts while maintaining robust temporal predictions (Jin et al., 2024)

## A.2 ADDITIONAL METHODS FOR GRAPH OOD GENERALIZATION

Beyond data augmentation, a growing body of work improves OOD generalization through model-based approaches that encode prior knowledge to learn stable, transferable representations. Representative methods include DisenGCN (Ma et al., 2019), IPGDN (Liu et al., 2020), FactorGCN (Yang et al., 2020), DisC (Fan et al., 2022), OOD-GNN (Li et al., 2022a), and CIGA (Chen et al., 2022). Disentanglement-based models such as DisenGCN and IPGDN separate latent factors using multi-channel convolutions and independence-promoting objectives. In contrast, causality-oriented methods like OOD-GNN and CIGA decorrelate causal and noncausal features or identify critical causal subgraphs to preserve stable relationships under distribution shifts.

Another major line of research focuses on learning-strategy methods, which refine training objectives and optimization schemes without altering the model architecture. Key directions include graph invariant learning (e.g., DIR (Wu et al., 2022c) and DIDA (Zhang et al., 2022)), which discovers invariant subgraphs or minimizes environment-wise risk; graph adversarial training (e.g., GraphAT (Feng et al., 2019) and WT-AWP (Wu et al., 2023)), which improves robustness through adversarial perturbations and co-adversarial optimization; and graph self-supervised learning (e.g., PATTERN (Yehudai et al., 2021) and RGCL (Li et al., 2022b)), which leverages contrastive or rationale-aware pretext tasks to learn generalizable representations. Together, these strategies complement data augmentation by enhancing stability and robustness across feature-level, topology-level, and hybrid distribution shifts.

Each category presents distinct trade-offs. Data augmentation is simple and broadly applicable, offering rapid robustness gains, but it may fail to cover truly novel distributions and can degrade performance if the augmentations diverge excessively from real data. Model-based methods provide strong theoretical grounding and capture stable causal or disentangled structures, yet they often require complex architectures and carefully chosen prior assumptions. Learning-strategy approaches are flexible and integrate easily with existing GNNs, but many rely on explicit or inferred environment splits, which limits effectiveness when such information is unavailable. Collectively, these methods are complementary and can be combined to achieve stronger OOD generalization.



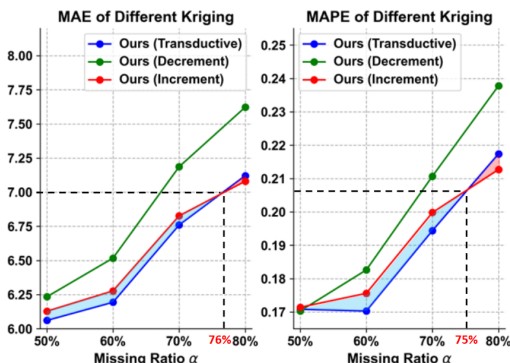

Figure 5: Comparison of decremental and incremental training strategies (Xu et al., 2025). The advantage of KITS (red line) becomes more pronounced as the missing ratio increases.

### A.3 INCREMENTAL AND DECREMENTAL TRAINING STRATEGY FOR INDUCTIVE KRIGING

Inductive kriging methods have traditionally been trained with a decremental strategy, in which values of some observed nodes are masked and the model learns to reconstruct them. KITS (Xu et al., 2025) reports that this approach produces a sparser training graph than the denser inference graph containing both observed and unobserved nodes, creating a "graph gap" that hampers transferability. To address this issue, KITS introduces an incremental training strategy that inserts virtual nodes during training to mimic future unobserved nodes and learns in a semi-supervised manner on the expanded graph, thereby aligning the topology of training and inference and improving generalization. These enhancements collectively reduce the graph-gap and the fitting issues that have limited previous inductive approaches. As shown in Figure 5, when the missing ratio is high and the graph is sparse, the insertion of virtual nodes further densifies the graph, making KITS's advantage even more pronounced.

## B THEORETICAL ANALYSIS OF DISTRIBUTION SHIFT IN INDUCTIVE KRIGING

**Goal.** Under the $3 \times 3$ setting, show that for any non-trivial family of normalized (space-time) graph convolutions $f_\theta$ (including the STGC in Eq. 1), training on the observed subgraph $\mathcal{G}_o = (V_o, \boldsymbol{A}_{oo})$ but evaluating on the enlarged graph $\mathcal{G} = (V_o \cup V_u, \boldsymbol{A})$ inevitably induces a distribution shift between training and testing (i.e., OOD), unless certain degenerate conditions hold.

**Preliminaries and notation.** Let $|V_o| = N_o$, $|V_u| = N_u > 0$. The adjacency is block-partitioned as in Eq. 2:

$$\boldsymbol{A} = \begin{pmatrix} \boldsymbol{A}_{oo} & \boldsymbol{A}_{ou} \\ \boldsymbol{A}_{uo} & \boldsymbol{A}_{uu} \end{pmatrix}, \qquad \boldsymbol{A}_{oo} = \boldsymbol{A}_o. \tag{16}$$

Define the degree matrix $\boldsymbol{D} = \operatorname{diag}(\boldsymbol{A}\boldsymbol{1})$ and the normalized propagation operator $\hat{\boldsymbol{A}} = \boldsymbol{D}^{-1/2}\boldsymbol{A}\boldsymbol{D}^{-1/2}$; write $\hat{\boldsymbol{A}}_{oo} = \boldsymbol{D}_{oo}^{-1/2}\boldsymbol{A}_{oo}\boldsymbol{D}_{oo}^{-1/2}$. The (space-time) propagation of the STGC stack (Eq. 1) is

$$\boldsymbol{H}^{(0)} = \boldsymbol{Z}_{i-m:i+m} \in \mathbb{R}^{N_o \times (2m+1)D}, \tag{17}$$

$$\boldsymbol{H}^{(l+1)} = \sigma\big(\hat{\boldsymbol{A}}\,\boldsymbol{H}^{(l)}\boldsymbol{W}_l\big), \quad l = 0, \dots, L-1, \tag{18}$$

where at training time $\hat{\boldsymbol{A}}$ is replaced by $\hat{\boldsymbol{A}}_{oo}$. From the Gaussian-process (GP) / kriging perspective, for spatial sets $V_o, V_u$ and a stationary kernel $k$,

$$\boldsymbol{K}_{oo} = [k(s_i, s_j)]_{i,j \in V_o}, \tag{19}$$

$$\boldsymbol{K}_{uo} = [k(s_i, s_j)]_{i \in V_u, j \in V_o}, \tag{20}$$

$$\boldsymbol{\mu}_{u|o} = \boldsymbol{K}_{uo}\boldsymbol{K}_{oo}^{-1}\boldsymbol{X}^o. \tag{21}$$

Empirical risk minimization in the main text reads

$$\mathcal{R}_{\text{train}}(\theta) = \mathbb{E}_{(\boldsymbol{X}^o, \boldsymbol{A}_{oo}) \sim P_o}\Big[\ell\big(f_\theta(\boldsymbol{X}^o; \boldsymbol{A}_{oo}), \boldsymbol{Y}^o\big)\Big], \tag{22}$$

$$\mathcal{R}_{\text{eval}}(\theta) = \mathbb{E}_{(\boldsymbol{X}^o, \boldsymbol{A}) \sim P_{ou}}\Big[\ell\big(f_\theta(\boldsymbol{X}^o; \boldsymbol{A}), \boldsymbol{Y}^u\big)\Big]. \tag{23}$$

**Proposition 1 (Structural shift: propagation operator changes).** If $N_u > 0$ and there exists at least one cross-block edge (some entry in $\boldsymbol{A}_{uo}$ or $\boldsymbol{A}_{ou}$ is positive), then $\hat{\boldsymbol{A}}$ and $\hat{\boldsymbol{A}}_{oo}$ have different spectra. Moreover, for any family with $\|\boldsymbol{W}_l\| > 0$ and non-constant activation $\sigma$, if $\boldsymbol{H}^{(0)}$ has non-zero covariance under the training distribution, then for some layer $l$ we have

$$\mathbb{P}_{\text{train}}(\boldsymbol{H}^{(l)}) \neq \mathbb{P}_{\text{test}}(\boldsymbol{H}^{(l)}). \tag{24}$$

*Proof.* Adding $V_u$ and cross-block edges modifies degrees from $\boldsymbol{D}_{oo}$ to

$$\boldsymbol{D} = \begin{pmatrix} \boldsymbol{D}_{oo} + \boldsymbol{A}_{ou}\boldsymbol{1} & 0 \\ 0 & \boldsymbol{D}_{uu} \end{pmatrix}, \tag{25}$$

so the *oo*-block of $\hat{\boldsymbol{A}} = \boldsymbol{D}^{-1/2}\boldsymbol{A}\boldsymbol{D}^{-1/2}$ differs from $\hat{\boldsymbol{A}}_{oo}$. If $\boldsymbol{A}_{ou} \neq 0$ or $\boldsymbol{A}_{uo} \neq 0$, standard matrix perturbation implies at least one eigenvalue shift, hence a spectral change. Applying different linear operators to identically distributed inputs, followed by a non-degenerate linear map $\boldsymbol{W}_l$ and non-constant $\sigma$, changes the output law; otherwise operator identifiability together with non-constancy of $\sigma$ would be violated. $\square$

**Proposition 2 (Target shift: supervision changes).** Training targets in-domain masked reconstruction on $V_o$ ($\boldsymbol{Y}^o$), while evaluation targets out-of-domain extrapolation on $V_u$ ($\boldsymbol{Y}^u$). If

$$\mathbb{P}\big(\boldsymbol{Y}^o \,\big|\, \boldsymbol{X}^o, \boldsymbol{A}_{oo}\big) \;\neq\; \mathbb{P}\big(\boldsymbol{Y}^u \,\big|\, \boldsymbol{X}^o, \boldsymbol{A}\big), \tag{26}$$

then the ERM solution $\theta^\star = \arg\min_\theta \mathcal{R}_{\text{train}}(\theta)$ generally does not minimize $\mathcal{R}_{\text{eval}}(\theta)$. *Proof.* The conditional laws differ because (i) the conditioning graph changes (Prop. 25), and (ii) the supervised index sets are disjoint ($V_o \cap V_u = \varnothing$), changing the support. For losses such as MAE/MSE, the risk minimizer is invariant across environments only if the two laws coincide or the problem degenerates (see Theorem below). $\square$

**Theorem (Inductive kriging is OOD except in degenerate cases).** Assume $N_u > 0$. Then the test distribution differs from the training distribution (i.e., OOD) unless one of the following degenerate situations holds:

1. **No cross-block edges:** $\boldsymbol{A}_{ou} = \boldsymbol{A}_{uo} = 0$, and the model at test time also uses only $\hat{\boldsymbol{A}}_{oo}$;
2. **Adjacency-invariant model:** every layer satisfies $\boldsymbol{W}_l = 0$ or $\sigma$ is constant, so outputs ignore $\hat{\boldsymbol{A}}$;
3. **Target equality:** $\mathbb{P}(\boldsymbol{Y}^u | \boldsymbol{X}^o, \boldsymbol{A}) = \mathbb{P}(\boldsymbol{Y}^o | \boldsymbol{X}^o, \boldsymbol{A}_{oo})$.

*Proof. Sufficiency:* Under (1), the evaluation operator equals the training operator and $V_u$ is unused; under (2), outputs are insensitive to adjacency; under (3), the target law is identical. Hence $\mathcal{R}_{\text{eval}} = \mathcal{R}_{\text{train}}$. *Necessity:* If any of (1)–(3) fails, then either Prop. 1 changes $\mathbb{P}(\boldsymbol{H}^{(l)})$ or Prop. 2 changes the target law; either implies $\mathcal{R}_{\text{eval}} \neq \mathcal{R}_{\text{train}}$. $\square$

**GP / kriging view (sufficient evidence for covariate shift).** Let the inter-node distance $d$ have different laws for "O–O pairs" vs. "U–O pairs": $p_{oo}(d) \neq p_{uo}(d)$. For a stationary kernel $k(d)$,

$$\mathbb{E}[\boldsymbol{K}_{oo}] = \mathbb{E}_{d \sim p_{oo}}[k(d)], \tag{27}$$

$$\mathbb{E}[\boldsymbol{K}_{uo}] = \mathbb{E}_{d \sim p_{uo}}[k(d)]. \tag{28}$$

If $p_{oo} \neq p_{uo}$ and $k$ is non-constant, the spectra and condition numbers of $\boldsymbol{K}_{oo}$ and $\boldsymbol{K}_{uo}$ generically differ; therefore the mapping

$$\boldsymbol{\mu}_{u|o} = \boldsymbol{K}_{uo}\boldsymbol{K}_{oo}^{-1}\boldsymbol{X}^o \tag{29}$$

at evaluation time is distributed differently from the masked estimator learned at training—constituting covariate shift. Only when $p_{oo} = p_{uo}$ or $k$ is constant (degenerate) do these differences vanish, matching the theorem.

Table 3: Overview of six datasets spanning three application domains.

| Data Type | Dataset | Data partition | | Region | Start Time | Samples | Nodes | Sampling Rate |
|---|---|---|---|---|---|---|---|---|
| | | Temporal | Spatial | | | | | |
| Traffic Speed | METR-LA | 7/1/2 | 3/1/1 | Los Angeles | 3/1/2012 | 34,272 | 207 | 5 minutes |
| | PEMS-BAY | 7/1/2 | 3/1/1 | San Francisco | 1/1/2017 | 52,116 | 325 | 5 minutes |
| Solar Power | NREL-AL | 7/1/2 | 3/1/1 | Alabama | 1/1/2016 | 105,120 | 137 | 5 minutes |
| | NREL-MD | 7/1/2 | 3/1/1 | Maryland | 1/1/2016 | 105,120 | 80 | 5 minutes |
| Air Quality (PM2.5) | AQI-36 | 1/1/1 | 3/1/1 | Beijing | 5/1/2014 | 8,759 | 36 | 1 hour |
| | AQI | 1/1/1 | 3/1/1 | 43 cities in China | 5/1/2014 | 8,760 | 437 | 1 hour |

**Conclusion.** Whenever previously unseen nodes participate in test-time propagation or supervision, inductive kriging is inherently an OOD problem. This aligns with the "OOD Problem" in the main text: the normalized operator changes from $\hat{A}_{oo}$ to $\hat{A}$ (degree & spectrum shift), and the target changes from $Y^o$ to $Y^u$. Consequently, intermediate features undergo both structural and covariate shifts, which the $3 \times 3$ split explicitly exposes; the robustness strategies in the method section can thus be viewed as targeted defenses against these shifts.

## C DETAILED EXPERIMENTAL SETUP

### C.1 DATASETS

In this appendix, we provide more details on the datasets that we used to run experiments. Table 3 summarizes the six datasets used in our experiments. Detailed descriptions are as follows:

- **METR-LA:** A traffic speed dataset containing average vehicle speeds from 207 detectors on Los Angeles County highways, collected every 5 minutes from March 1 to June 27, 2012.
- **PEMS-BAY:** A traffic speed dataset comprising measurements from 325 sensors in the San Francisco Bay Area, sampled every 5 minutes between January 1 and June 30, 2017.
- **NREL-AL:** A solar power dataset recording output from 137 photovoltaic plants in Alabama throughout 2016, with 5-minute sampling intervals.
- **NREL-MD:** A solar power dataset capturing output from 80 photovoltaic plants in Maryland during 2016, sampled every 5 minutes.
- **AQI-36:** A subset of the Air Quality Index (AQI) dataset selected from the Urban Computing Project, beginning on May 1, 2014, and commonly used in kriging studies.
- **AQI:** The full Air Quality Index dataset containing hourly measurements of six pollutants from 437 monitoring stations across 43 Chinese cities; consistent with prior work such as GRIN (Cini et al., 2022) and KITS (Xu et al., 2025), we focus on PM2.5 concentrations.

**Data partition.** To ensure fair evaluation and consistent comparison across methods, we partition each dataset along both temporal and spatial dimensions. For all datasets except AQI-36 and AQI, we adopt a temporal split of 7:1:2 for training, validation, and testing, respectively. In parallel, we apply a spatial split of 3:1:1, dividing the monitoring locations (e.g., sensors, photovoltaic plants, or stations) into three groups for training, validation, and testing. For the AQI-36 and AQI datasets, which span longer periods and exhibit pronounced seasonal patterns, we follow KITS (Xu et al., 2025) and adopt a temporal split of 1:1:1 to capture seasonal variability. Entire months are allocated to each subset: March, June, September, and December form the test set, February, May, August, and November constitute the validation set, and the remaining months are used for training. The spatial split remains 3:1:1, dividing monitoring nodes into training, validation, and testing groups as in the other datasets. This dual partitioning strategy—uniformly separating data across time and space—encourages models to generalize to unseen periods and locations, providing a rigorous assessment of forecasting performance.

**Creating Random Missing.** For most experiments, the missing ratio $\alpha$ is fixed at 25% across all datasets. To create random missing patterns, we shuffle the node order and partition by index using a fixed spatial split of 3:1:1 for training, validation, and testing, respectively. Specifically, suppose a dataset contains $N$ nodes and the missing ratio is $\alpha = 0.25$. We first generate a random permutation

of all node indices to eliminate any spatial or temporal bias. Based on this shuffled sequence, we assign the first $\lfloor 0.6N \rfloor$ nodes to the **training set**, the next $\lfloor 0.2N \rfloor$ nodes to the **validation set**, and the remaining nodes to the **test set**. To ensure reproducibility, we fix the random seed to 42 for both the numpy and random libraries across all datasets by default.

**Data Normalization.** Proper normalization is essential to stabilize model training and ensure comparability across heterogeneous measurements. We consider two widely used approaches:min–max normalization (implemented via the MinMaxScaler in scikit-learn) and zero-mean normalization (implemented via the StandardScaler in scikit-learn). For the NREL-AL and NREL-MD solar power datasets, the rated capacity (i.e., maximum output) of each photovoltaic plant is known. We apply min–max normalization on a per-node basis, dividing each node's time series by its own capacity so that the normalized values lie in the range $[0, 1]$. This node-specific scaling preserves the relative generation profile of each plant while removing the effect of differing absolute capacities. For all other datasets, where node-level maximum values are either unavailable or not meaningful (e.g., traffic speed or air-quality measurements), we uniformly adopt zero-mean normalization, transforming each feature to have zero mean and unit variance. This standardization balances the input scale across variables and facilitates stable, efficient training for downstream models.

**Constructing Adjacency Matrix.** A widely used approach for constructing a spatial adjacency matrix is to apply a thresholded Gaussian kernel, which connects each node to its geographically nearby nodes within a specified radius. The weighted adjacency matrix is defined as

$$\boldsymbol{A}(v, u) = \exp\big( - \|\mathbf{s}_v - \mathbf{s}_u\|^2 / \sigma^2 \big) \cdot \mathbf{1}\{\|\mathbf{s}_v - \mathbf{s}_u\| \leq \delta\}, \qquad v, u \in V, \tag{30}$$

where $\boldsymbol{A}$ is the adjacency matrix; $v$ and $u$ are node indices; $\mathbf{s}_v$ and $\mathbf{s}_u$ denote the spatial coordinates of nodes $v$ and $u$; $\sigma > 0$ is the Gaussian kernel bandwidth (length scale) controlling how rapidly the edge weight decays with distance; and $\delta > 0$ is the distance threshold (radius) that sparsifies the graph by retaining only nearby connections. Intuitively, the exponential term assigns higher weights to edges linking spatially closer nodes, while the indicator function $\mathbf{1}\{\|\mathbf{s}_v - \mathbf{s}_u\| \leq \delta\}$ ensures that long-range connections beyond the threshold $\delta$ are removed, yielding a sparse graph that reflects local spatial correlations. Following GRIN (Cini et al., 2022) and KITS (Xu et al., 2025), we set $\sigma$ to the empirical standard deviation of all pairwise node distances, which provides a data-driven scale for the Gaussian kernel and avoids manual tuning.

## C.2 BASELINES

**Mean Imputation.** Missing values are filled with the average of all available node observations at each time interval, rather than node-wise means, to avoid bias from sparsely observed nodes.

**OKriging.** Ordinary Kriging exploits the geographic relationships among nodes and models spatial correlations with a Gaussian process to perform purely spatial interpolation.

**KNN.** The K-Nearest Neighbors method estimates the value of an unobserved node by averaging the values of its ten nearest neighbors (K = 10) based on geographic distance.

**KCN (Appleby et al., 2020).** KCN first unifies GCNs and kriging by directly using neighbor observations within the convolution—recovering classical kriging as a special case—and augments it with attention for better interpolation. Among its three variants—Graph Convolutional Networks (GCN), Graph Attention Networks (GAT), and GraphSAGE—we evaluate the GraphSAGE implementation.

**IGNNK (Wu et al., 2021a).** IGNNK learns a transferable spatial message-passing scheme via random subgraphs and signal reconstruction, enabling inductive kriging on unseen sensors/graphs. It Uses a total-reconstruction loss over all nodes (not only masked ones), encouraging global generalization of message passing.

**INCREASE (Zheng et al., 2023).** INCREASE encodes three heterogeneous relations—spatial proximity, functional similarity, and transition probability—and uses relation-aware GRUs plus multi-relation attention to fuse spatiotemporal signals for inductive kriging at new locations.

**KITS (Xu et al., 2025).** KITS bridges the train–inference 'graph gap' by incrementally adding virtual nodes during training, pairing/fusing them with similar observed nodes and supervising with pseudo labels, so the learned patterns transfer reliably to real unobserved nodes.

Table 4: Error bars of inductive methods with 4 different random seeds on the METR-LA dataset. Seed 0 corresponds to 42, seed 1 corresponds to 3407, seed 2 corresponds to 1202, seed 3 corresponds to 6666. The best results are shown in **bold**, and the second-best results are underlined. "Improvements" indicate the performance gain of our DRIK method over the best baseline.

| Method | Metric | Seed 0 | Seed 1 | Seed 2 | Seed 3 | Mean ± Std |
|---|---|---|---|---|---|---|
| Mean | | 8.272 | 9.318 | 8.282 | 8.932 | 8.701±0.473 |
| OKriging | | 7.294 | 7.907 | 7.793 | 7.663 | 7.664±0.239 |
| KNN | | 7.987 | 8.332 | 8.881 | 8.610 | 8.453±0.374 |
| KCN | | 7.190 | 7.281 | 8.112 | 7.101 | 7.421±0.455 |
| IGNNK | MAE | 5.801 | 6.006 | 7.579 | 6.479 | 6.466±0.726 |
| INCREASE | | 5.992 | 6.221 | 7.680 | 6.978 | 6.718±0.732 |
| KITS | | 5.666 | 6.031 | 6.848 | 6.621 | 6.292±0.475 |
| DRIK(Ours) | | **5.197** | **5.450** | **6.731** | **5.635** | **5.753±0.603** |
| Improvements | | **8.280%** | **9.260%** | **1.710%** | **14.890%** | **8.035%** |
| Mean | | 11.417 | 12.804 | 11.334 | 12.845 | 12.100±0.748 |
| OKriging | | 10.277 | 11.354 | 10.871 | 11.299 | 10.950±0.429 |
| KNN | | 12.370 | 13.151 | 13.423 | 13.901 | 13.211±0.563 |
| KCN | | 12.470 | 12.490 | 13.110 | 13.371 | 12.860±0.376 |
| IGNNK | RMSE | 8.914 | 9.686 | 11.311 | 10.562 | 10.118±0.955 |
| INCREASE | | 9.198 | 10.095 | 12.120 | 11.624 | 10.759±1.157 |
| KITS | | 8.981 | 9.945 | 10.257 | 11.043 | 10.057±0.810 |
| DRIK(Ours) | | **8.101** | **8.895** | **9.763** | **8.955** | **8.929±0.583** |
| Improvements | | **9.120%** | **8.160%** | **4.820%** | **15.210%** | **9.328%** |
| Mean | | 22.133 | 27.922 | 22.677 | 29.392 | 25.031±3.298 |
| OKriging | | 18.896 | 23.822 | 20.528 | 24.610 | 21.964±2.393 |
| KNN | | 19.820 | 25.307 | 22.728 | 26.487 | 23.585±2.904 |
| KCN | | 23.983 | 24.583 | 23.556 | 24.181 | 24.076±0.427 |
| IGNNK | MAPE | 15.581 | 18.313 | 18.945 | 21.864 | 18.676±2.366 |
| INCREASE | | 16.854 | 18.494 | 18.960 | 25.345 | 19.913±3.350 |
| KITS | | 15.096 | 18.230 | 17.648 | 21.832 | 18.202±2.712 |
| DRIK(Ours) | | **13.154** | **15.740** | **17.043** | **16.722** | **15.665±1.520** |
| Improvements | | **12.860%** | **13.660%** | **3.430%** | **23.410%** | **13.840%** |

## C.3 EVALUATION METRICS

We mainly adopt Mean Absolute Error (MAE), Mean Absolute Percentage Error (MAPE), and Mean Relative Error (MRE) to evaluate the performance of all methods. The formulas are given as follows:

$$MAE = \frac{1}{|\Omega|} \sum_{i \in \Omega} \left| \boldsymbol{Y}_i - \hat{\boldsymbol{Y}}_i \right| \tag{31}$$

$$RMSE = \sqrt{\frac{1}{|\Omega|} \sum_{i \in \Omega} \left( \boldsymbol{Y}_i - \hat{\boldsymbol{Y}}_i \right)^2} \tag{32}$$

$$MAPE = \frac{1}{|\Omega|} \sum_{i \in \Omega} \frac{\left| \boldsymbol{Y}_i - \hat{\boldsymbol{Y}}_i \right|}{\left| \boldsymbol{Y}_i \right|} \tag{33}$$

where $\Omega$ is the index set of unobserved nodes used for evaluation, $\boldsymbol{Y}$ denotes the ground-truth data, $\hat{\boldsymbol{Y}}$ is the estimation generated by the kriging models, and $\bar{\boldsymbol{Y}}$ is the average value of the labels.

## C.4 IMPLEMENTATION DETAILS FOR REPRODUCIBILITY

Our code is implemented in Python 3.8 with PyTorch 1.8.1, PyTorch Lightning 1.4.0, and CUDA 11.3. All experiments are conducted on a single NVIDIA A100 80GB GPU. Unless otherwise noted, we fix the random seed of numpy, random, PyTorch, and PyTorch Lightning to 42,

Table 5: Error bars of inductive methods with 4 different random seeds on the PEMS-BAY dataset. Seed 0 corresponds to 42, seed 1 corresponds to 3407, seed 2 corresponds to 1202, seed 3 corresponds to 6666. The best results are shown in **bold**, and the second-best results are underlined. "Improvements" indicate the performance gain of our DRIK method over the best baseline.

| Method | Metric | Seed 0 | Seed 1 | Seed 2 | Seed 3 | Mean $\pm$ Std |
|---|---|---|---|---|---|---|
| Mean | | 4.999 | 4.916 | 4.896 | 4.654 | 4.866$\pm$0.147 |
| OKriging | | 4.874 | 4.887 | 4.874 | 4.609 | 4.811$\pm$0.124 |
| KNN | | 5.678 | 5.678 | 5.628 | 5.320 | 5.576$\pm$0.162 |
| KCN | | 4.676 | 4.779 | 4.693 | 4.559 | 4.677$\pm$0.085 |
| IGNNK | MAE | 3.445 | 3.593 | 3.556 | 3.392 | 3.497$\pm$0.083 |
| INCREASE | | 3.599 | 3.804 | 3.770 | 3.494 | 3.667$\pm$0.130 |
| KITS | | 3.410 | 3.651 | 3.733 | 3.590 | 3.596$\pm$0.132 |
| DRIK(Ours) | | **3.218** | **3.468** | **3.419** | **3.254** | **3.340$\pm$0.105** |
| Improvements | | **5.630%** | **3.490%** | **3.850%** | **4.070%** | **4.260%** |
| Mean | | 8.474 | 8.366 | 8.243 | 7.775 | 8.215$\pm$0.269 |
| OKriging | | 8.266 | 8.351 | 8.108 | 7.717 | 8.111$\pm$0.252 |
| KNN | | 10.431 | 10.347 | 9.941 | 9.682 | 10.100$\pm$0.342 |
| KCN | | 9.253 | 10.110 | 9.198 | 8.015 | 9.144$\pm$0.838 |
| IGNNK | RMSE | 6.067 | 6.248 | 6.126 | 5.876 | 6.079$\pm$0.138 |
| INCREASE | | 6.850 | 6.886 | 6.878 | 6.228 | 6.711$\pm$0.309 |
| KITS | | 6.445 | 6.521 | 6.638 | 6.279 | 6.471$\pm$0.149 |
| DRIK(Ours) | | **5.840** | **6.009** | **5.919** | **5.550** | **5.830$\pm$0.193** |
| Improvements | | **3.750%** | **3.840%** | **3.380%** | **5.540%** | **4.130%** |
| Mean | | 12.862 | 12.267 | 11.979 | 11.069 | 12.044$\pm$0.712 |
| OKriging | | 12.412 | 12.027 | 11.732 | 10.855 | 11.757$\pm$0.607 |
| KNN | | 14.087 | 13.371 | 13.044 | 12.107 | 13.152$\pm$0.887 |
| KCN | | 13.514 | 14.011 | 13.010 | 12.827 | 13.341$\pm$0.509 |
| IGNNK | MAPE | 8.378 | 8.577 | 8.268 | 7.592 | 8.204$\pm$0.362 |
| INCREASE | | 9.457 | 9.748 | 9.536 | 8.211 | 9.238$\pm$0.631 |
| KITS | | 8.602 | 8.775 | 8.733 | 8.092 | 8.550$\pm$0.300 |
| DRIK(Ours) | | **7.728** | **8.101** | **7.561** | **7.310** | **7.675$\pm$0.296** |
| Improvements | | **7.760%** | **5.560%** | **8.550%** | **3.720%** | **6.900%** |

and set the missing ratio to $\alpha = 25\%$. For all datasets, following KITS (Xu et al., 2025), the temporal window size is 24, and the feature dimension and batch size are fixed at 64 and 32, respectively. Within the STGC module, the parameter $m$ is set to 1, indicating that spatio-temporal feature aggregation uses data from one historical, one current, and one future time interval. We employ the Adam optimizer with a fixed learning rate of 0.0001 and apply gradient clipping with a threshold of 1.0 to stabilize training. The model is trained for up to 300 epochs with an early stopping strategy: validation is performed after each epoch, and training halts if the validation performance shows no improvement for 50 consecutive epochs. The model achieving the best validation performance is saved and used for final inference.

# D    MORE EXPERIMENTAL RESULTS

## D.1    MODEL STABILITY UNDER DIFFERENT NODE DIVISIONS

To assess how each method performs when the set of observed and unobserved nodes varies, we evaluate model stability across different node divisions. For each dataset with a fixed missing ratio $\alpha$, we randomly partition nodes into training, validation, and test groups using four different random seeds, producing distinct spatial splits and missing patterns. Each method is trained and tested on these splits, and we report the mean and standard deviation of key metrics in Tables 4, 5, and 6. This analysis captures two aspects of stability: (1) the model's ability to learn consistently despite random parameter initialization, and (2) its robustness to changes in the spatial distribution of observed

Table 6: Error bars of inductive methods with 4 different random seeds on the AQI-36 dataset. Seed 0 corresponds to 42, seed 1 corresponds to 3407, seed 2 corresponds to 1202, seed 3 corresponds to 6666. The best results are shown in **bold**, and the second-best results are underlined. "Improvements" indicate the performance gain of our DRIK method over the best baseline.

| Method | Metric | Seed 0 | Seed 1 | Seed 2 | Seed 3 | Mean ± Std |
|---|---|---|---|---|---|---|
| Mean | | 18.431 | 16.800 | 24.983 | 24.021 | 21.059±3.50 |
| OKriging | | 16.003 | 14.824 | 21.105 | 20.920 | 18.713±2.88 |
| KNN | | 14.727 | 13.517 | 20.420 | 22.122 | 17.697±3.89 |
| KCN | | 21.963 | 14.111 | 19.978 | 20.019 | 19.018±3.27 |
| IGNNK | MAE | 20.138 | 13.683 | 18.055 | 18.272 | 17.037±2.88 |
| INCREASE | | 16.963 | 12.437 | 17.411 | 17.150 | 15.990±2.24 |
| KITS | | 19.600 | 13.531 | 18.831 | 19.294 | 17.814±2.78 |
| DRIK(Ours) | | **13.443** | **10.949** | **15.991** | **16.502** | **14.221±2.32** |
| Improvements | | **8.710%** | **11.970%** | **8.160%** | **3.780%** | **8.655%** |
| Mean | | 31.631 | 27.266 | 42.812 | 42.233 | 35.985±7.02 |
| OKriging | | 28.744 | 24.879 | 37.603 | 37.401 | 32.157±6.26 |
| KNN | | 26.800 | 23.300 | 37.425 | 39.620 | 31.286±6.82 |
| KCN | | 36.647 | 25.510 | 38.579 | 36.778 | 34.378±5.93 |
| IGNNK | RMSE | 33.993 | 21.483 | 31.361 | 31.586 | 29.606±5.58 |
| INCREASE | | 32.854 | 22.126 | 32.948 | 31.159 | 29.772±4.90 |
| KITS | | 34.668 | 23.851 | 36.738 | 37.055 | 33.078±6.22 |
| DRIK(Ours) | | **25.550** | **18.260** | **29.513** | **29.345** | **25.667±4.58** |
| Improvements | | **4.670%** | **15.000%** | **5.890%** | **5.820%** | **7.845%** |
| Mean | | 49.586 | 51.848 | 85.004 | 65.456 | 62.974±14.7 |
| OKriging | | 42.670 | 46.733 | 71.152 | 61.158 | 55.428±10.7 |
| KNN | | 37.737 | 42.213 | 66.196 | 61.294 | 51.860±12.8 |
| KCN | | 57.988 | 45.123 | 59.001 | 59.333 | 55.361±6.55 |
| IGNNK | MAPE | 69.964 | 42.215 | 56.094 | 48.399 | 54.668±10.4 |
| INCREASE | | 41.619 | 32.662 | 46.929 | 42.331 | 40.885±5.79 |
| KITS | | 76.466 | 33.981 | 48.896 | 37.327 | 49.168±16.5 |
| DRIK(Ours) | | **28.433** | **30.018** | **46.054** | **37.090** | **35.399±7.50** |
| Improvements | | **24.650%** | **8.100%** | **1.860%** | **0.630%** | **8.810%** |

versus unobserved nodes. The results reveal how each inductive kriging approach maintains—or loses—performance when node divisions vary. From Tables 4, 5, and 6, we observe the following:

- **DRIK consistently outperforms all baseline methods across all datasets and metrics.** For instance, on METR-LA (Table 4), DRIK achieves an average MAE of $5.753 \pm 0.603$, which is $8.035\%$ lower than the best baseline (KITS: $6.292 \pm 0.475$). Similarly, on PEMS-BAY (Table 5), DRIK reduces MAE by $4.26\%$ compared to KITS, and on AQI-36 (Table 6), it achieves an $8.655\%$ improvement in MAE over INCREASE. These gains are consistent across RMSE and MAPE, demonstrating the robustness of DRIK's three-tier strategy.

- **DRIK exhibits strong stability under varying node divisions.** The standard deviations of DRIK's metrics are competitive and often lower than those of other methods. For example, on METR-LA, DRIK's MAE standard deviation is $0.603$, compared to $0.475$ for KITS and higher values for other baselines. On AQI-36, DRIK's MAE standard deviation is $2.32$, which is lower than most baselines, indicating consistent performance despite changes in node composition.

- **The impact of randomness due to parameter initialization and node division is well mitigated by DRIK.** The relatively small standard deviations across runs suggest that DRIK is less sensitive to initial conditions and spatial splits. This stability is particularly notable in complex scenarios such as AQI-36, where seasonal and spatial heterogeneity are pronounced.

In summary, DRIK not only achieves superior predictive accuracy but also maintains robust performance under different node divisions, highlighting its suitability for real-world inductive kriging applications where sensor layouts may vary.

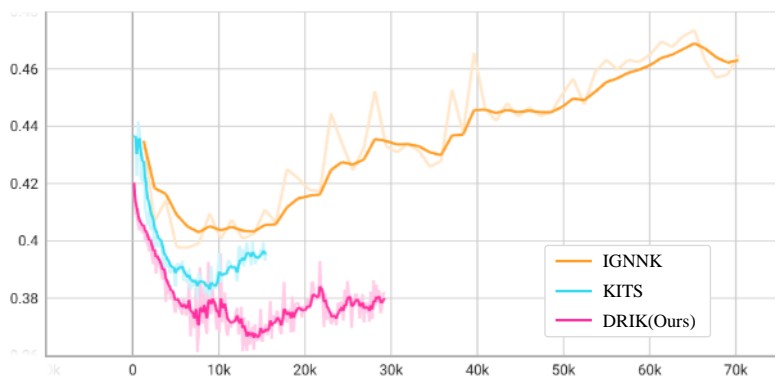

Figure 6: Comparison of validation loss during training across methods.

## D.2 VISUALIZATION OF LOSS CURVE DURING TRAINING

This section provides a qualitative view of the training dynamics of inductive kriging methods, complementing the quantitative results. Visualizing loss curves (Figure 6) reveals how each model converges, whether it overfits or underfits, and how stable learning remains across different initializations and node splits—key factors for assessing robustness under the proposed 3×3 partitioning scheme.

In Figure 6, training lengths differ intentionally: slower models were extended to observe full convergence and potential overfitting. This enables a clearer comparison of learning trends. DRIK shows smoother, more stable convergence than IGNNK and KITS, with a steadily decreasing validation loss that plateaus without rebound, indicating lower sensitivity to noise and distribution shifts. It also reaches a lower validation loss more consistently across random seeds and node divisions, aligning with its design to enhance robustness through node perturbation, edge dropping, and subgraph addition.

Overall, the loss curves highlight DRIK's training stability and resistance to overfitting, supporting its strong OOD generalization.

