# OpenReview forum: "DRIK: Distribution-Robust Inductive Kriging without Information Leakage"
_ICLR.cc/2026/Conference — ICLR 2026 Conference Withdrawn Submission_

### Official Review · Reviewer_JRV7 · 2025-10-18

**Soundness:** 2
**Presentation:** 3
**Contribution:** 3
**Rating:** 6
**Confidence:** 2

**Summary:**

This paper proposes DRIK (Distribution-Robust Inductive Kriging), a framework for spatio-temporal interpolation under distribution shifts. The authors identify two major challenges in existing inductive kriging methods—information leakage caused by improper spatio-temporal splits and poor generalization under unseen spatial distributions. To address these, they introduce a 3×3 spatial-temporal data partitioning scheme to prevent leakage and a three-layer robustness strategy—node perturbation, edge dropping, and subgraph addition to enhance distribution robustness. Extensive experiments on six real-world datasets demonstrate that DRIK consistently outperforms baseline models in interpolation accuracy and out-of-distribution generalization.

**Strengths:**

1) The problem is well-motivated. This paper provides a clear diagnosis of information leakage in previous GNN-based kriging studies and introduces a principled 3×3 data partitioning strategy.

2) The proposed method is technically solid. The three-layer (node–edge–subgraph) robustness mechanisms are intuitively designed. They address spatial perturbations, topological uncertainties, and unseen node adaptation in a unified manner.

3) The experiments results shows the effectiness of the proposed method.

**Weaknesses:**

(1) It would be better to elaborate more on the definition of OOD in this paper. In the introduction the author mentioned "Under the new setting, the key out-of-distribution (OOD) property of inductive kriging becomes clear". From the spatial perspective, distribution at different location can be different, which is also referred as spatial heterogeneity. From the temporal perspective, OOD also means  temporal non-stationarity or evolving patterns.

(2) Following the first point, this paper seems to pay more attention on the ``spatial'' OOD, while pays less attention to the temporal OOD which is also quite important in the spatial-temporal task, especially when the author claim the proposed method is "DISTRIBUTION-ROBUST"

(3) Another concerns is on the complexity of the proposed method,  the  two-pass kriging process in subgraph addition and the use of expanded graphs can increase training complexity.

**Questions:**

please see the weakness above.

---

### Official Review · Reviewer_xfby · 2025-10-22

**Soundness:** 2
**Presentation:** 4
**Contribution:** 1
**Rating:** 2
**Confidence:** 5

**Summary:**

This paper works on graph-based spatio-temporal inductive kriging, where some nodes are unknown during training, yet would be dynamically inserted into the original graph for value estimation during inference stage. This paper reveals the issue of information leakage in previous kriging baselines, raised by the fact that validation and test subset share the same set of nodes, and previous methods would use early stop on validation subset to obtain a good model "fitting" the test subset. To address this issue, this paper differentiate the sets of nodes for validation and testing, and focuses on the setting of out-of-domain (OOD) kriging.

**Strengths:**

1. The motivation of using distinct sets of nodes for validation and testing is reasonable.
2. Extensive experiments have been conducted.
3. The reported performance is good.

**Weaknesses:**

1. The technical contribution of this paper is quite weak:
  - I find the methodology part is highly similar to that of KITS (Xu et al., 2025), e.g., STGC and Subgraph Addition parts.
  - Node perturbation and edge dropping are 2 widely accepted data augmentation approaches.
2. In Table 2, SA would degrade the performance much, but can improve the performance when other modules are introduced, a detailed case study is recommended here to provide some insights about the use of this module.
3. KITS randomly insert new nodes during training, while in Subgraph Addition, new nodes preserved for validation are inserted during training, there are no comparisons between these two strategies to compare their effectiveness and show their pros and cons.
  - Besides, in Figure 4 and Obs 8, when the missing ratio is high, DRIK is not as good as KITS, the current explanation in Obs 8 is unclear and not so convincing, please explain these two strategies in detail.
4. Since this paper focuses on the OOD setting, can you follow KITS to experiment with the cases when a kriging model is trained on one dataset (e.g., METR-LA) and directly tested on another dataset (e.g., PEMS-BAY)?

**Questions:**

N.A.

---

### Official Review · Reviewer_5emE · 2025-10-27

**Soundness:** 4
**Presentation:** 4
**Contribution:** 3
**Rating:** 4
**Confidence:** 3

**Summary:**

This paper deals with the information leakage and OOD generalization issues in inductive kriging problem. The authors propose to avoid information leakage by decoupling data split across temporal and spatial dimensions. And a three-layer strategy named DRIK is introduced to improve the OOD generalization. DRIK is consist of three designs including node perturbation, task-aware edge dropping, and subgraph addition. Extensive experiments on six datasets show the improvements on generalization and validity of the proposed DRIK strategies.

**Strengths:**

**Originality** This work demonstrates original contributions in two folds: (1) raised the issue in traditional settings which do not fully isolate data in temporal and spatial dimensions for training/validation/testing sets, and proposed a redefined setting to avoid information leakage (2) Three-layer strategies are dedicatedly design for inductive kriging problem by considering the distinctive graph characteristics.

**Quality** & **Clarity** This work is well written with clear structure. Specially, the methodology components and contributions are clearly presented. Moreover, extensive experiments, ablation studies, and comprehensive appendix materials provide strong support for this work.

**Weaknesses:**

1. Lots of previous works have similar designs on graph data augmentations as mentioned in section 2, and the authors claim that the proposed three-layer augmentation designs differ from these works by considering the distinctive graph characteristics. But these related augmentation methods (like GAug, KDGA, FLAG, GREA, etc. mentioned in Line 153- Line 156) are not quantitatively compared in experiments.

2. In figure 3, DRIK demonstrates worse validation MAE than IGNNK and KITS on NERL-MD dataset, which contradicts the conclusion that DRIK has stronger generalization performance.

3. Unclear how many nodes are randomly selected in $\mathcal{M}_1$ for Eq. (12) for subgraph addition, and how will the size of $\mathcal{M}_1$ and $\mathcal{M}_2$ influence the overall performance.

4. Edge dropping function with different masks in Eq. (12) and Eq.(14) should be clearly distinguished.

**Questions:**

Several questions are raised in "Weaknesses" part. The main concern is the lack of quantitative comparisons with most related "graph data augmentation" works. I would be willing to change my recommendation according to the authors' response.

---

### Note · Authors · 2025-11-14

I have read and agree with the venue's withdrawal policy on behalf of myself and my co-authors.